# Effects of Ultrasonic-Assisted Extraction on the Physicochemical Properties of Different Walnut Proteins

**DOI:** 10.3390/molecules24234260

**Published:** 2019-11-22

**Authors:** Siyi Lv, Ahmed Taha, Hao Hu, Qi Lu, Siyi Pan

**Affiliations:** 1College of Food Science and Technology, Huazhong Agricultural University, Wuhan 430070, China; 15927085695@163.com (S.L.); ahmed-taha@alexu.edu.eg (A.T.); huhao@mail.hzau.edu.cn (H.H.); 2Center for Evaluation and Inspection of Hubei Food and Drug Administration, Wuhan 430070, China; 3Department of Food Science, Faculty of Agriculture (Saba Basha), Alexandria University, Alexandria 21531, Egypt; 4Research Institute of Agricultural Products Processing and Nuclear-Agricultural Technology, Hubei Academy of Agricultural Sciences, Wuhan 430070, China

**Keywords:** walnut proteins, albumin, globulin, glutelin, ultrasonic-assisted extraction, physicochemical properties

## Abstract

The effects of ultrasonic-assisted extraction (UAE, 200 W, 20 min) on the yield and physicochemical properties of different walnut proteins (WNPs, including albumin, globulin, and glutelin) were investigated. Sodium dodecyl sulfate polyacrylamide gel electrophoresis (SDS-PAGE) analysis indicated that UAE could result in protein molecular fragmentation of albumin, but did not affect the major bands of globulin and glutelin. The CD spectra demonstrated that different WNPs obtained by UAE had different changes in their secondary structure. Under UAE, there was an increase in surface hydrophobicity (*H*_0_) of albumin and gluten and no change in the fluorescence intensity, while decreases were observed in the *H_0_* and fluorescence intensity of globulin; and the contents of total and surface free sulfhydryl in albumin dramatically decreased. UAE reduced the size of the particles and the dimension of the microstructures in albumin and gluten, indicating that ultrasound could unfold protein aggregates. In addition, UAE increased the solubility, emulsifying activity (EA), foaming capacity (FC), and foam stability (FS) of the obtained proteins. The above results indicate that ultrasound extraction is a promising approach to improve the extraction yield and properties of walnut proteins.

## 1. Introduction

Walnut is considered to be a nutritious food with an anti-brain aging effect and the ability to lower total plasma cholesterol and low-density lipoprotein [1,2]. Walnut contains up to 68% walnut oil [3], which is rich in monounsaturated and polyunsaturated fatty acids [4], as well as many ingredients that are beneficial to human health [5], such as vitamins, arginine, folic acid, cellulose, tannins, and polyphenols [6]. Walnut proteins (WNPs), which account for 18%–24% of the walnut’s total mass, are the main byproduct of walnut oil processing [7]. There are four major categories of walnut proteins: glutelin, globulin, albumin, and prolamin, which respectively account for 70.11%, 17.57%, 6.81%, and 5.33% of the total walnut protein content [8,9]. However, the low solubility and extraction rate of walnut proteins largely hinder their utilization, resulting in a major waste of protein resources.

Ultrasound methods are widely applied in food processing [10], such as extraction [11], defoaming, enzyme inactivation, and sterilization [12,13]. Moreover, in our recent studies, ultrasound was used as a promising emulsification technique to prepare protein emulsions [14,15,16]. Furthermore, compared with traditional methods, ultrasonic-assisted extraction (UAE) is considered as an effective extraction method that can greatly reduce the use of solvents and extraction time [17]. UAE can increase the extraction yield of proteins from defatted rice bran [18], perilla [19], defatted soy flakes [20], and the sea bass [21]. In general, ultrasound can produce strong shear forces, disrupt the cell structure [22], and increase the mass and heat transfer [23,24], which can enhance the yield of extracted components and the extraction rate.

The low solubility of walnut proteins [9] severely restrains their functions in many foods, and thus physicochemical modifications may help to improve their functional properties. As a new approach, high-intensity ultrasound could either increase protein yield or improve protein properties when used in protein extraction. Krešić et al. [25] showed that ultrasound treatment of whey protein (20 kHz) could greatly improve the apparent viscosity and solubility of the protein. Madadlou et al. [26] found that ultrasound treatment of casein suspension could significantly reduce the turbidity; in addition, they also proposed that sonication (130 and 24 kHz) could delay the gelation point of casein gels to a lower pH and increase the elasticity of casein gels [27]. Zisu et al. [28] revealed that ultrasonic processing (4 kW) in large scale reactors could reduce the viscosity of whey protein and casein while increasing their gel strength. The improvement of the physicochemical properties of proteins by UAE may be due to the cavitation effect, which can cause sudden rupture of the protein and the production of microfluidics, leading to the exposure of some molecular structures to the solvent.

However, little is known about the extraction efficiency of UAE on different components of walnut proteins, and its effects on the protein physicochemical properties under the same extraction conditions. It was hypothesized that UAE could reduce the extraction time and increase the yield of walnut proteins. Moreover, we investigated the differences in structural and functional properties of walnut albumin, globulin, and gluten under the same extraction conditions. The results may provide a better understanding of the effects of UAE and its applications in the field of food proteins.

## 2. Materials and Methods

### 2.1. Materials

Walnuts were purchased from a local market (Xinjiang, China) and defatted by repeated extraction with hexane until clarification could occur according to the method of Wolf et al. [29] with minor modifications. The defatted walnuts were air-dried in the fume hood overnight to remove residual solvent. After crushing, 25-mesh defatted walnut powders were collected, which were then stored in valve bags at −20 °C until use.

Sodium dodecyl sulfate (SDS), 5,5′-dithiobis (2-nitrobenzoic acid) (DTNB), tris, anilino-8-naphthalene sulfonate (ANS), β-mercaptoethanol, glycine and urea were purchased from Sigma Adrich Company (St. Louis, MO, USA). All other reagents were purchased from the Yuanye Biotechnology Company (Shanghai, China).

### 2.2. Ultrasonic-Assisted Extraction of Walnut Proteins

Different WNPs (albumin, globulin, gluten) were prepared by the modification of previously studied methods [9,30]. The defatted walnut flour (1:30 *w*/*v*) was dispersed in distilled water for constant magnetic stirring (traditional extraction method, as the control) and ultrasonic treatment. An ultrasonic cell disruptor (JY 92-ΙΙ DN, NingBo Scientz Co. Ltd., Zhejiang, China) with a 0.636 cm diameter titanium probe was operated to sonicate 30 mL of walnut protein slurry in a 50-mL centrifuge tube inserted into the ice-water bath. Albumin was subsequently extracted under the conditions summarized in Table 1. Then, the slurry was centrifuged at 10,000× *g* for 30 min at 4 °C and the precipitate collected from the constant magnetic stirring step was used for the next extraction of walnut proteins. The precipitates were sequentially dispersed in 1.0 M NaCl and 0.1 M NaOH, and UAE of globulin and gluten was carried out using the above method. The concentration of the collected supernatant was detected with Coomassie Brilliant Blue method using a bovine serum albumin standard curve [31]. The supernatant was adjusted to pH 8.0 using 0.5 M HCl, dialyzed against deionized water for 48 h, and then lyophilized. The purity of different WNPs was measured by the AOAC standard [32]. The yield of different WNPs was calculated by the following formula: Protein yield (%) = (mass of supernatant protein/mass of total protein)(1)

### 2.3. Determination of Ultrasonic Time and Power Output

The power output values of ultrasonic treatment were measured by a power meter (Shenzhen Northmeter Co., Ltd., Shenzhen, China). The Energy density was estimated by the following equation of Koh et al. [33] and the values were shown in Table 1. Determination of ultrasonic time and power output was based on the highest total protein yield.

Energy density (J/mL) = Power output (W) × Time(s)/Volume (mL) (2)

### 2.4. Effect of Ultrasonic-Assisted Extraction on Structural Properties

#### 2.4.1. SDS-PAGE

The molecular weight of WNPs from the control and UAE was analyzed by reducing and non-reducing sodium dodecyl sulfate polyacrylamide gel electrophoresis (SDS-PAGE) using 16% separating gel and 8% stacking gel according to previous studies with some modifications [34,35]. The protein powders were mixed in 0.1 M Tris–HCl buffer (pH = 6.8) containing 5% β-mercaptoethanol (the reducing conditions), 2% SDS, 0.02% bromophenol blue and 10% glycerin, with sample concentration of 2 mg/mL. The mixed solutions were heated at boiling temperature for 5 min, while 15 µL of the prepared samples were injected into each well and then ran to the bottom of the gel at 100 V. The gel was stained using the Coomassie Blue (R-250) staining method and the molecular weight was determined by comparison with the MW standard markers (BIO-RAD) (MW 10.0~180.0 kDa). Images were analyzed by Imgae Lab software version 5.0 (2000 Alfred Nobel Drive, Bio-Rad Laboratories Inc., WDC, USA).

#### 2.4.2. Secondary Structure

The sample solutions (0.1 mg/mL in 0.01 M phosphate buffer, pH 7.0) of WNPs from the control and UAE were centrifuged at 10,000× *g* for 30 min at 4 °C, and the Circular dichroism (CD) spectra (1500-150, Jasco Corp., Tokyo, Japan) were collected in the wavelength range of 190–260 nm. The bandwidth, response time, and scan rate were 1.0 nm, 0.25 s, and 100 nm/min under nitrogen flux with a 0.1 cm path length quartz cell at ambient temperature (25 ± 1 °C) [36]. The secondary structure (α-helix, β-sheet, β-turn, and random coil) of WNPs was estimated by the Yang-Us, jwr software (1500-150, Jasco Corp., Tokyo, Japan).

#### 2.4.3. Surface Hydrophobicity

The surface hydrophobicity (*H*_0_) of WNPs of the control and UAE was determined according to the previously described method [37] with minor modifications by using 1-anilino-8-naphthalene sulfonate (ANS) as a hydrophobic fluorescence probe. Briefly, protein samples were prepared in 0.01 M sodium phosphate buffer (pH 7.0) to achieve concentrations of 0.0005–0.1 mg/mL. ANS solution of 60 mL (8.0 mM in 0.01 M phosphate buffer, pH 7.0) was added to 3 mL of the sample solution. The fluorescence intensity was measured with the excitation and emission wavelengths of 360 and 480 nm at ambient temperature by a fluorescence spectrophotometer (F-4600, Hitachi, Tokyo, Japan). The initial slope of the fluorescence intensity versus sample concentration (mg/mL) plots was calculated by linear regression analysis as an index of *H*_0_.

#### 2.4.4. Contents of Total and Surface Free Sulfhydryl 

Contents of total and surface free sulfhydryl of WNPs from the control and UAE were determined using Ellman’s reagent (5,5′-dithiobis (2-nitrobenzoic acid), DTNB) using the previously described method [38,39] with minor modifications. Protein samples were mixed in standard buffer (pH = 8.0) containing 0.086 M Tris, 0.09 M glycine, and 4 mM Na_2_EDTA for surface free sulfhydryl, and in denaturing buffer containing standard buffer plus 0.5% sodium dodecyl sulfate and 6 M urea for total sulfhydryl. The protein solutions were adjusted to 2 mg/mL and incubated in a shaking water bath at room temperature for 24 h, and then centrifuged at 10,000× *g* at 4 °C for 30 min. The supernatant of 3 mL was added with 0.03 mL Ellman’s reagent solution (4 mg/mL DTNB in standard buffer) and then rapidly mixed and allowed to stand at 25 °C for 15 min. After the absorbance was read at 412 nm on UV spectrophotometer (UV-1700, Shimatzu, Japan) with the subtraction of the blank (0.03 mL Ellman’s reagent solution in buffers), the contents of total and surface free sulfhydryl were calculated using the molar extinction coefficient of 1.36 × 10^4^ M^−1^ cm^−1^.

#### 2.4.5. Intrinsic Fluorescence

The intrinsic fluorescence emission spectra of WNPs from the control and UAE (0.2 mg/mL in 0.01 M phosphate buffer, pH 7.0) were obtained using a fluorescence spectrophotometer (F-4600, Hitachi, Tokyo, Japan) with a 1-cm path length cell at room temperature (25 ± 1 °C). The fluorescence emission spectrum was determined according to a previous method [40] with minor modifications at the excitation wavelength of 290 nm, emission spectra between 300 and 400 nm, and a scanning speed of 240 nm/min. The data of all fluorescence experiments were the average of three scans.

#### 2.4.6. Particle Size Distribution

The particle size distribution of WNPs from the control and UAE (*w*/*v*, 2% in distilled water) was examined by static light scattering using a particle sizing instrument (Mastersizer 2000, Malvern Instruments Ltd., Worcestershire, UK), with a refractive index of 1.330, absorption parameter of 0.001, and the pump speed of 2000 rpm. The particle size was expressed as the average of three readings of the surface-weighted mean diameter (D_3,2_) and the volume-weighted mean diameter (D_4,3_).

#### 2.4.7. Scanning Electron Microscopy 

The morphology of WNPs from the control and UAE was observed using a scanning electron microscopy (SEM) (JSM-IT 300, Tokyo, Japan) at an acceleration voltage of 20 kV. Before using SEM, the samples were coated with gold in the argon atmosphere.

### 2.5. Effect of Ultrasonic-Assisted Extraction on Functional Properties

#### 2.5.1. Protein Solubility 

The solubility of WNPs from the control and UAE was determined using the method of Hu et al. with minor modifications [41]. Protein solutions (*w*/*v*, 2% in distilled water) were centrifuged at 10,000× *g* for 30 min at 4 °C. The protein concentration in the supernatant was analyzed by the Bradford method using BSA as a standard. Protein solubility was calculated as the percentage of supernatant protein content/total protein content in the sample. All results were the average of three determinations.

#### 2.5.2. Emulsifying Activity (EA) and Emulsion Stability (ES)

Emulsifying properties of WNPs from the control and UAE were determined using the method of Tsumura et al. [42] with some modifications. 20 mL of protein solution (*w*/*v*, 2% in distilled water) and 5 mL of soybean oil were mixed together, and then homogenized at 20,000 rpm/min for 2 min using a homogenizer (T 18, IKA, Germany). The emulsions (50 μL) were immediately pipetted out from the bottom after homogenization at 0 and 10 min, and diluted with 10 mL of SDS solution (*w*/*v*, 1% in distilled water). The absorbance of the solutions was recorded at 500 nm using a UV spectrophotometer (UV-1700, Shimatzu, Japan). EA and ES values were calculated as follows:EA (m^2^/g) = (2 × 2.303 × absorbance at 0 min × dilution factor/protein concentration × oil volume fraction × 10,000) (3)

ES (min) = (absorbance at 0 min/absorbance at 0 min − absorbance at 10 min) × 10(4)

#### 2.5.3. Foaming Capacity (FC) and Foam Stability (FS)

The foaming properties of WNPs from the control and UAE were determined using the method of Jain et al. [43] with minor modifications. 50 mL of protein solution (*w*/*v*, 2% in distilled water) was homogenized at 20,000 rpm/min for 2 min with a homogenizer (T 18, IKA, Germany), and then transferred into a 100-mL measuring cylinders. The volume of foam was measured at 0 and 30 min. FC and FS values were calculated as follows:FC (%) = (Volume after homogenizing − Initial volume)/Initial volume

FS (%) = (Volume after standing − Initial volume)/Initial volume.

### 2.6. Statistical Analysis

All samples were analyzed in triplicate. SPSS 23.0 software was used for analysis of variance (ANOVA). One-way ANOVA test and Duncan’s test were used for the analysis of significant differences (*p* < 0.05).

## 3. Results and Discussion

### 3.1. Determination of Conditions for Ultrasonic-Assisted Extraction

The effects of ultrasonic power and time on the extraction efficiency of WNPs were investigated. As clearly shown in Figure 1, ultrasound as an assisted means could significantly increase the extraction rate with the total yields of 50.75% (different ultrasonic powers) and 54.19% (different ultrasonic times), respectively. This effect might be due to the turbulence and shear force produced by UAE, which could effectively destroy the molecular structures of proteins in order to form soluble proteins [44,45]. However, long-time (30 min) and high-power (400 W, 500 W at 176, 220 J/mL) ultrasonic extraction reduced the yield of WNPs. One possible reason for these results was the formation of protein aggregates, which was further confirmed by the results of the subsequent studies of the particle size (Table 2) and surface hydrophobicity (Figure 2). Therefore, the optimal ultrasonic power and ultrasonic time were determined to be 200 W and 20 min based on the highest total yield of walnut proteins, respectively.

### 3.2. Effect of Ultrasonic-Assisted Extraction on Structural Properties

#### 3.2.1. SDS-PAGE

As shown in Figure 3, for WNPs from the control and UAE under non-reducing and reducing conditions, UAE did not cause obvious changes in the protein electropherogram. Under reducing conditions (Figure 3A), no differences in the electrophoretic profiles were observed between the WNPs from the control and those from UAE. This result is consistent with the results of O’Sullivan et al. [46,47] and Higuera-Barraza et al. [48] in the studies of animal and vegetable proteins (20 kHz, 34 W cm^−2^ and 2 min), wheat and soy protein isolates (20 kHz, 34 W cm^−2^ and 2 min), and squid mantle proteins (20 kHz, 20, and 40% amplitude).

Under non-reducing conditions (Figure 3B), five strong bands appeared for WNPs, with molecular distributions ranging from 40–55 kDa, 35–40 kDa, 25–35 kDa, 15–25 kDa, and <15 kDa. For the albumin obtained by UAE, there was one less molecule in the molecular weight range of 15–25 kDa, while the band intensity of 40–55 kDa was remarkably increased, suggesting that UAE could cause the fragmentation of protein molecules to form disulfide bonds. This result was also supported by the contents of total and surface free sulfhydryl (Figure 4). Nevertheless, the main bands for the globulin and glutelin obtained by UAE did not change significantly. Similar observations were reported by Zhu et al. [36] for walnut protein isolate (600 W and 30 min). The differences in molecular distribution after UAE between these proteins might be due to the differences in protein type.

There were some differences between the reducing and non-reducing conditions. The band intensity of the molecular weight range < 15 kDa increased under reducing conditions. This result might be due to the fact that β-mercaptoethanol destroyed the disulfide bond between protein molecules, resulting in the appearance of smaller molecules.

#### 3.2.2. Secondary Structure

The CD spectra of the WNPs from the control and UAE are shown in Table 3. There were significant changes in the secondary structure of the proteins extracted by UAE. Compared with those of the control sample, α-helix was increased by 10.84% while β-sheet was decreased by 9.96% in sample B (albumin). Similar results were reported from studies in which ultrasound was used to treat whey protein concentrate (20 kHz, 450 W and 60 min) and bovine serum albumin (20 W cm^−2^ and 45 min) [49,50]. In sample D (globulin), a-helix and β-sheet were increased by 3.3% and 10.47%, while β-turn and random coil were decreased by 6.1% and 7.67%, respectively. These findings were in agreement with those reported for soy protein isolate (400 W and 15 min) [37]. However, in sample F (glutelin), a-helix and β-sheet were decreased by 2.10% and 4.34%, while β-turn and random coil were increased by 5.37% and 1.04%, respectively. The different changes in the secondary structure of these proteins might be due to different protein structures. This is because the secondary structure of the protein depends not only on intermolecular and intramolecular interactions, but also on the local sequences of the amino acids.

#### 3.2.3. Surface Hydrophobicity

As can be seen from Figure 2, gluten in walnut protein has a higher surface hydrophobicity than albumin and globulin. Albumin and gluten obtained by UAE also had the highest *H*_0_, probably due to the cavitation phenomenon induced by ultrasonic treatment that exposed some buried hydrophobic sites of the protein to the surface. This finding is consistent with a previous report [51,52] on zein and glutelin (600 W and 40 min), and wheat germ protein (600 W and 30 min). However, *H*_0_ decreased after ultrasonic extraction for globulin, indicating the occurrence of protein aggregation that protected the hydrophobic regions of the protein. A similar conclusion was drawn by Jiang et al. [53] and Chandrapala et al. [49] for black bean protein isolates (450 W and 60 min) and whey protein concentrate (450 W and 60 min). Ultrasound might result in protein denaturation, which might lead to more extensive intermolecular interactions and entrapped hydrophobic sites. These results are also supported by the data of particle size (Table 2).

#### 3.2.4. Total and Surface Free Sulfhydryl 

The changes in the contents of total and surface free sulfhydryl of WNPs from the control and UAE are shown in Figure 4. No significant changes in the contents of total and surface free sulfhydryl were observed for globulin and glutelin. Similar results were obtained for soy protein isolate and egg white protein (20 kHz, 20 min) [54]. However, the contents of total and surface free sulfhydryl in albumin dramatically decreased, possibly because the sulfhydryl groups located in the interior of the molecules were exposed to protein aggregates by S–S bonds under ultrasound treatment. This was in agreement with the SDS-PAGE results (Figure 3B) and was consistent with previous studies by Gülseren et al. [50] of bovine serum albumin (20 W cm^−2^ and 45 min), and Hu et al. [37] of soybean protein isolate (20 kHz and 400 W). Furthermore, the inconsistent results for various proteins may be ascribed to the fact that ultrasound may have little effect on the free SH groups in globulin and gluten due to the complexity of different protein fractions and intermolecular positions of the SH groups.

#### 3.2.5. Intrinsic Fluorescence

The conformational changes that lead to microenvironmental changes around the tyrosine and phenylalanine residues in WNPs of the control and UAE were examined by the emission fluorescence spectra (Figure 5). The maximum emission fluorescence intensity of the glutelin and albumin (excited at 280 nm) of the control was found at 343.2 nm and 334.6 nm, while the maximum wavelength of glutelin and albumin obtained by UAE shifted to 350.6 nm and 337.0 nm, respectively. This red shift was due to molecular unfolding that resulted in an increase in the polarity of more chromophores. However, fluorescence intensity decreased for globulin obtained by UAE, suggesting the occurrence of protein aggregation. A similar result where ultrasound decreased the fluorescence intensity of ovalbumin was also reported [55]. 

#### 3.2.6. Particle Size Distribution

The effects of UAE on the particle size distribution, D_4,3_ and D_3,2_, are shown in Figure 6 and Table 2. D_4,3_ is able to monitor the changes in aggregates, while D_3,2_ is highly representative of the size of most particles [54]. For albumin and gluten obtained by UAE, the particle size distribution was relatively narrow, and the mean particle diameter decreased significantly. D_4,3_ in albumin decreased from around 26.24 μm to around 18.57 μm and in gluten from around 157.50 μm to around 138.28 μm. These data indicate that the larger insoluble aggregates in the samples might be destroyed by the cavitation, turbulence, and shear force of the ultrasound, resulting in smaller particles. Contrarily, globulin obtained by UAE presented higher peak values and particle size than that from the control sample, suggesting the ultrasound-induced formation of protein aggregates. Similarly, Hu et al. [37] and Gülseren et al. [50] have also found an increase in D_4,3_ and D_3,2_, which suggests the formation of aggregates. As shown above by the electrophoretic profiles (Figure 3B), no change was observed in the molecular distribution. The formation of aggregates might not be attributed to the covalent bonds between protein molecules such as disulfide bridges. In contrast, noncovalent interactions, such as hydrophobic and electrostatic interactions, might be the driving force.

#### 3.2.7. Scanning Electron Microscopy

The microstructures of WNPs from the control and UAE were observed by scanning electron microscopy. Figure 7 shows a set of SEM images at the magnification of 300. Dimensions of the SEM images for the albumin and gluten from UAE (Figure 7B,F) appeared to be smaller than those of the control (Figure 7A,E). The cavitation and turbulent force caused by UAE might reduce the D_3,2_ of samples (Table 2) to result in the formation of uniform and dense structure. However, the image of the globulin from UAE (Figure 7D) showed larger dimensions than that of the control (Figure 7C), suggesting the formation of aggregates. One possible reason for the formation of aggregates was that UAE facilitated protein unfolding and exposed the hydrophobic parts (Figure 2) and free sulfhydryl (Figure 4A), which could interact with each other and lead to the formation of aggregates [56].

### 3.3. Effect of Ultrasound-Assisted Extraction on the Functional Properties

#### 3.3.1. Solubility Property

The degree of solubility determines the production and processing properties of proteins such as emulsification and gelation. As shown in Figure 8, the solubility of WNPs obtained by UAE was significantly higher than that of the control. Similar results were obtained for the solubility of meat protein [57] and soy protein isolate [37] after ultrasound treatment. On the one hand, the increase in solubility may be due to the decrease in the particle size of proteins (Table 2) and the increase in protein-water interactions [52]. On the other hand, the increase in solubility might result from the formation of soluble protein aggregates by the internal hydrophilic exposure and the conformational change [58].

#### 3.3.2. Emulsifying Property

The emulsifying property of proteins is an important indicator in food processing, such as beverage production, cake making, and seasoning preparation. EA and ES of WNPs are shown in Figure 9. The EA of WNPs extracted by UAE was greatly improved, while the ES was reduced. The results were consistent with the reports that long-time and high-power ultrasound could improve the emulsifying properties of peanut proteins [59] and walnut protein isolate [36], possibly because the partial variability and disordered structure of the protein under ultrasonic extraction would lead to better adsorption of the protein at the oil-water interface. Meanwhile, the insoluble protein aggregates produced due to UAE are not conducive to maintaining the stability of oil-water interface.

#### 3.3.3. Foaming Property

Foaming capacity and foam stability of WNPs obtained by UAE were improved (Figure 10). FC and FS were increased significantly for the albumin by 20.03% and 9.54%, and for the glutelin by 14.1% and 10.58%, and the increase was the highest for the globulin by 49.39% and 11.21%. Foaming power of whey protein and rapeseed protein was similar to that reported by Jambrak et al. [60] and Dong et al. [61]. The increase in foaming property was probably due to the cavitation and shearing forces of UAE, which disperse the protein particles more evenly and improve the foaming property. UAE might result in partially unfolded structures of proteins that facilitate their absorption at the oil-water interface, which is accompanied by an increase in foaming power. Therefore, WNPs could be used as a good foaming agent for baked foods, desserts, and ice cream. 

## 4. Conclusions

UEA could be considered as a useful extraction method that could increase the extraction yield, change the structural properties, and improve the solubility, EA, FC, and FS of walnut proteins. However, the structural properties of WNPs vary with the types of proteins under the same ultrasonic time and power. After UAE, the globulin *H*_0_ and fluorescence intensity decreased, and the particle size and microscopic size became larger, indicating the appearance of protein aggregates. Interestingly, the cavitation and shear forces induced by UAE interrupted the molecular structure of the albumin, thereby forming disulfide bonds. In addition, the functional properties of the gluten were significantly improved by UAE. UAE might be significant for achieving industrial production of WNPs and utilization of functional components of natural plant proteins.

## Figures and Tables

**Figure 1 molecules-24-04260-f001:**
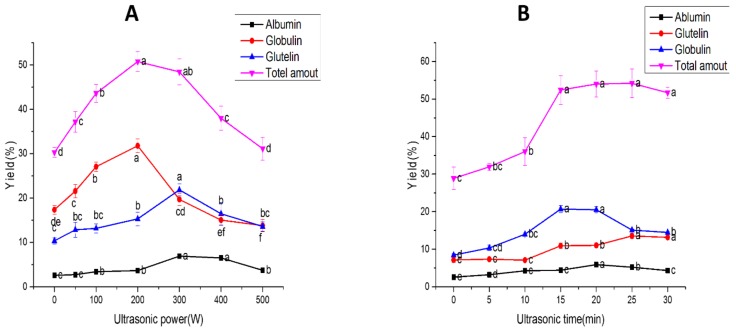
Yield of walnut proteins (WNPs): (**A**) different ultrasonic powers at 20 min ultrasonic time, (**B**) different ultrasonic times at 200 W ultrasonic power.

**Figure 2 molecules-24-04260-f002:**
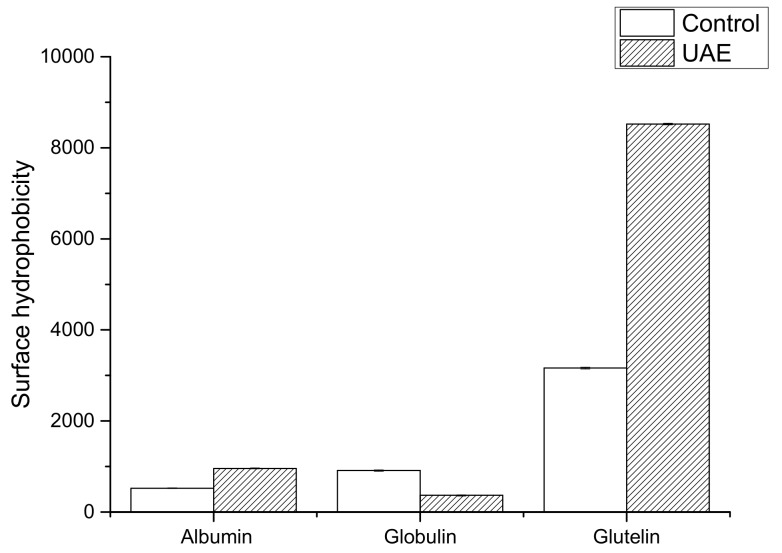
Surface hydrophobicity of WNPs from the control and ultrasonic-assisted extraction (200 W and 20 min).

**Figure 3 molecules-24-04260-f003:**
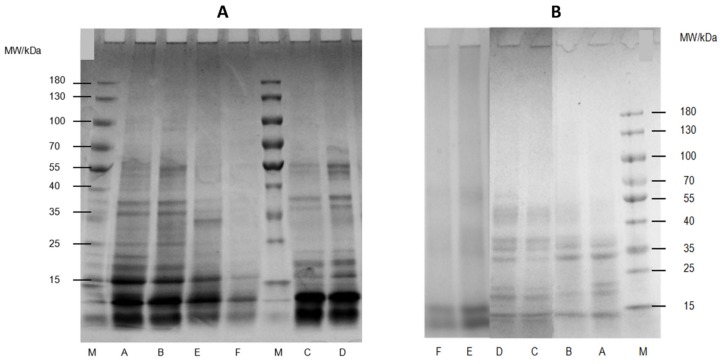
SDS-PAGE electrophoretic profiles of WNPs from the control and ultrasonic-assisted extraction (200 W and 20 min) under reducing (Figure 3A) and non-reducing (Figure 3B) conditions: (**A**) albumin of the control, (**B**) albumin of UAE, (**C**) globulin of the control, (**D**) globulin of UAE, (**E**) glutelin of the control, (**F**) glutelin of UAE, (M) MW standard markers.

**Figure 4 molecules-24-04260-f004:**
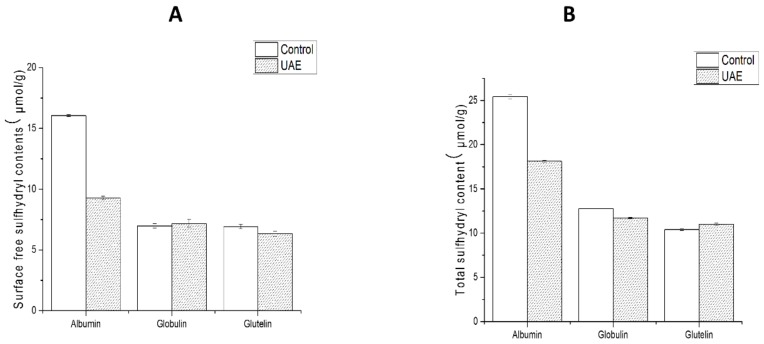
Contents of total and surface free sulfhydryl of WNPs from the control and ultrasonic-assisted extraction (200 W and 20 min): (**A**) Content of surface free sulfhydryl, (**B**) Content of total sulfhydryl.

**Figure 5 molecules-24-04260-f005:**
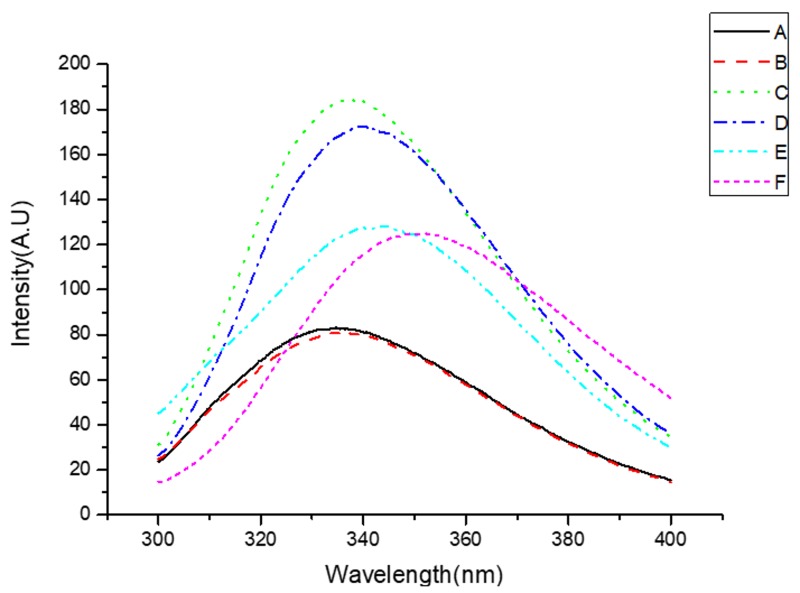
Fluorescence emission spectra of WNPs from the control and ultrasonic-assisted extraction (200 W and 20 min): (**A**) albumin of the control, (**B**) albumin of UAE, (**C**) globulin of the control, (**D**) globulin of UAE, (**E**) glutelin of the control, (**F**) glutelin of UAE.

**Figure 6 molecules-24-04260-f006:**
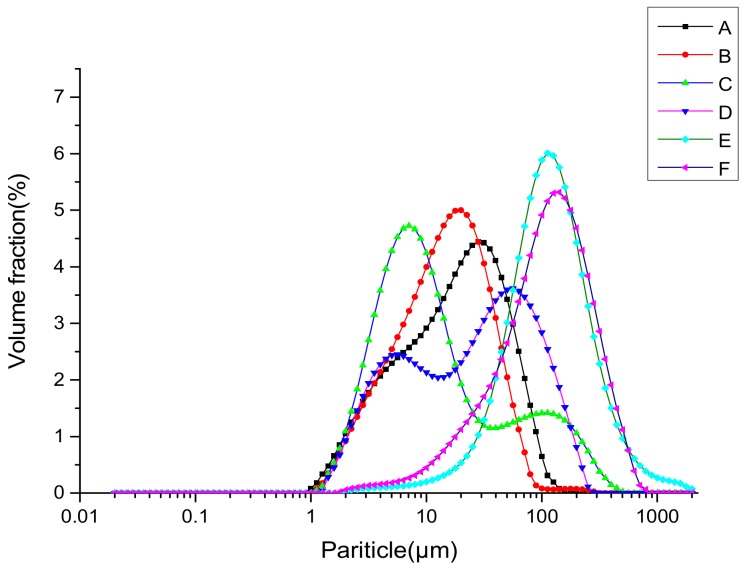
Particle size distribution of WNPs from the control and ultrasonic-assisted extraction (200 W and 20 min): (**A**) albumin of the control, (**B**) albumin of UAE, (**C**) globulin of the control, (**D**) globulin of UAE, (**E**) glutelin of the control, (**F**) glutelin of UAE.

**Figure 7 molecules-24-04260-f007:**
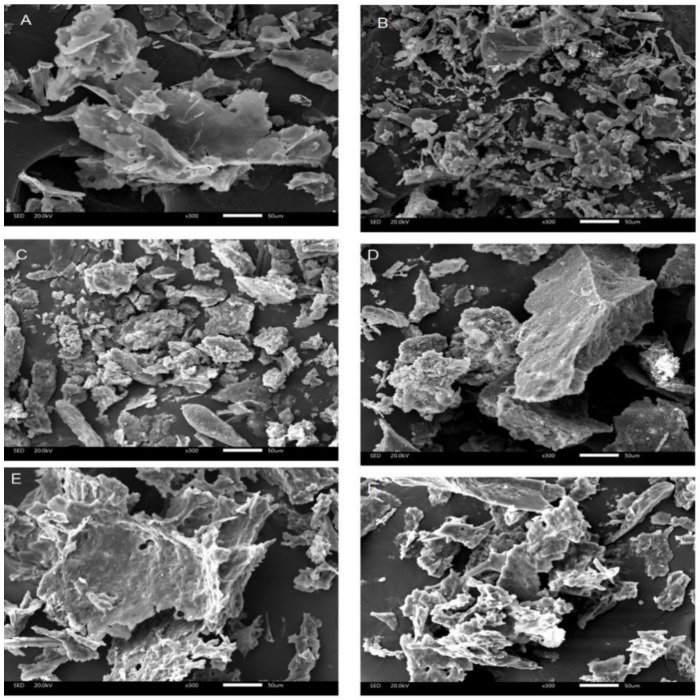
SEM images of WNPs from the control and ultrasonic-assisted extraction (200 W and 20 min): (**A**) albumin of the control, (**B**) albumin of UAE, (**C**) globulin of the control, (**D**) globulin of UAE, (**E**) glutelin of the control, (**F**) glutelin of UAE.

**Figure 8 molecules-24-04260-f008:**
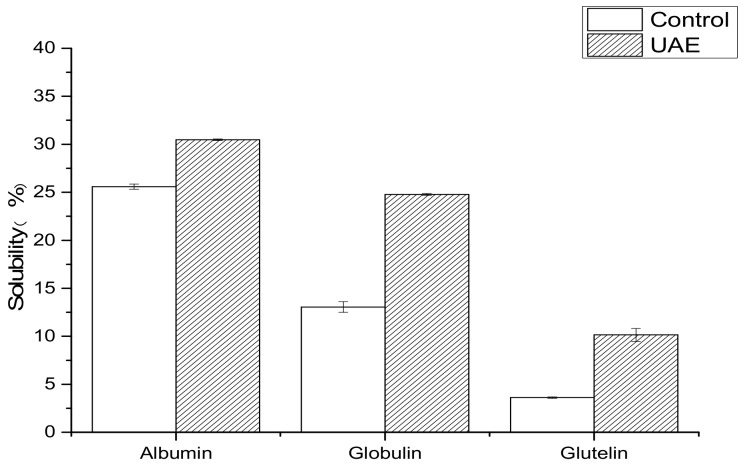
Solubility of WNPs from the control and ultrasonic-assisted extraction (200 W and 20 min).

**Figure 9 molecules-24-04260-f009:**
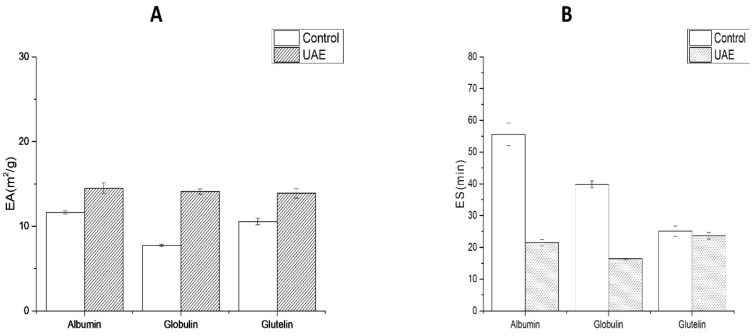
Emulsifying activity (EA) and emulsion stability (ES) of WNPs from the control and ultrasonic-assisted extraction (200 W and 20 min): (**A**) EA, (**B**) ES.

**Figure 10 molecules-24-04260-f010:**
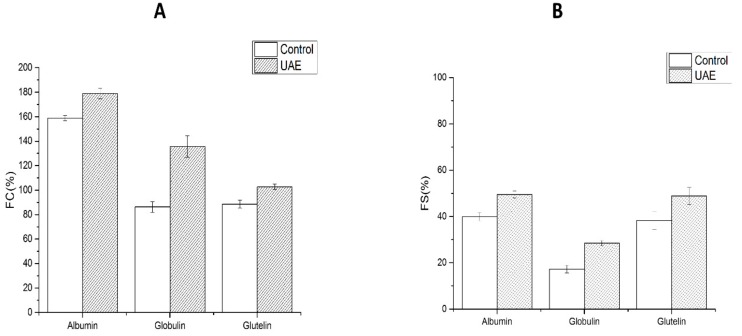
Foaming capacity (FC) and foam stability (FS) of WNPs from the control and ultrasonic-assisted extraction (200 W and 20 min): (**A**) FC, (**B**) FS.

**Table 1 molecules-24-04260-t001:** Conditions of ultrasonic-assisted extraction.

Ultrasonic Extraction	Conditions	Energy Density (J/mL)
Diameter titanium probe (cm)	0.636	/
Frequency (kHz)	20	/
Power output (W)	0, 100, 200, 300, 400, 500	0, 44, 88, 132, 176, 220
Time (min)	0, 5, 10, 15, 20, 25	0, 22, 44, 66, 88, 110
pulse duration (s)	on-time 4/off-time 2	/

**Table 2 molecules-24-04260-t002:** Volume-weighted mean diameter (D43) and the surface-weighted mean diameter (D32) of WNPs from the control and ultrasonic-assisted extraction (200 W and 20 min).

Samples	D_43_ (μm)	D_32_ (μm)
A (albumin of the control)	26.24 ± 0.22	9.38 ± 0.02
B (albumin of UAE)	18.57 ± 0.19	8.21 ± 0.01
C (globulin of the control)	33.59 ± 0.58	6.88 ± 0.02
D (globulin of UAE)	40.89 ± 0.23	9.70 ± 0.02
E (glutelin of the control)	157.50 ± 1.06	63.33 ± 0.04
F (glutelin of UAE)	138.28 ± 0.99	48.62 ± 0.33

**Table 3 molecules-24-04260-t003:** Secondary structure of WNPs from the control and ultrasonic-assisted extraction (200 W and 20 min).

Samples	α-Helix (%)	β-Sheet (%)	β-Turn (%)	Random Coil (%)
A (albumin of the control)	15.73 ± 0.21	55.83 ± 0.11	0.1 ± 0.02	28.30 ± 0.11
B (albumin of UAE)	26.57 ± 0.26	45.87 ± 0.50	0 ± 0.0	27.60 ± 0.24
C (globulin of the control)	16.63 ± 0.14	11.63 ± 0.46	27.77 ± 0.26	43.97 ± 0.12
D (globulin of UAE)	19.93 ± 0.32	22.10 ± 0.33	21.67 ± 0.52	36.30 ± 0.13
E (glutelin of the control)	2.10 ± 0.07	43.67 ± 0.57	6.5 ± 0.28	47.73 ± 0.45
F (glutelin of UAE)	0 ± 0	39.33 ± 0.09	11.87 ± 0.06	48.77 ± 0.07

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
