# Peer review of "Effects of Ultrasonic-Assisted Extraction on the Physicochemical Properties of Different Walnut Proteins"

_molecules, 2019, doi:10.3390/molecules24234260_

Round 1
Reviewer 1 Report
The work is methodical. Its purpose is to examine the optimal conditions for the extraction of organic substances from walnuts. It may be useful for other researchers. Before publishing, please explain two things:
Describe the energy parameters of the process in more detail. Input energy says little. I do not understand the description contained in lines 83-87: 30 ml of the sample was tested, 0.6 cm electrode and 10,000 g of matter? This continues later in the textAuthor Response
Please see the attachment

Reviewer 2 Report
The backround of the research is interesting and significant from the scientific point of view. Also, all materials and methods are adequately described and presented. Results are well presented and all tables and graphs are necessary to show a main achievements of study. Discussion follows the results which are adequately compared with other relevant scientific data and researches. Conclusions support obtained results and in a concise and clear way respond to the aim of the research.
Round 2
Reviewer 1 Report
I accept changes.